# The gammaherpesviral TATA-box-binding protein directly interacts with the CTD of host RNA Pol II to direct late gene transcription

Angelica F. Castañeda[1◎], Allison L. Didychuk[1◎], Robert K. Louder[2,3¤], Chloe O. McCollum[4], Zoe H. Davis[5], Eva Nogales[2,4,6,7], Britt A. Glaunsinger[1,4,6,7]*

1 Department of Plant and Microbial Biology, University of California, Berkeley, CA, United States of America, 2 Molecular Biophysics and Integrative Bio-Imaging Division, Lawrence Berkeley National Laboratory, Berkeley, CA, United States of America, 3 Biophysics Graduate Group, University of California, Berkeley, CA, United States of America, 4 Department of Molecular and Cell Biology, University of California Berkeley, CA, United States of America, 5 Division of Infectious Diseases and Immunity, School of Public Health, University of California, Berkeley, CA, United States of America, 6 California Institute for Quantitative Biosciences (QB3), University of California, Berkeley, CA, United States of America, 7 Howard Hughes Medical Institute, Berkeley, CA, United States of America

◎ These authors contributed equally to this work.
¤ Current address: Department of Biology, Johns Hopkins University, Baltimore, MD, United States of America
* glaunsinger@berkeley.edu

**Data Availability Statement:** All relevant data are within the manuscript and its Supporting Information files.

## Abstract

β- and γ-herpesviruses include the oncogenic human viruses Kaposi's sarcoma-associated virus (KSHV) and Epstein-Barr virus (EBV), and human cytomegalovirus (HCMV), which is a significant cause of congenital disease. Near the end of their replication cycle, these viruses transcribe their late genes in a manner distinct from host transcription. Late gene transcription requires six virally encoded proteins, one of which is a functional mimic of host TATA-box-binding protein (TBP) that is also involved in recruitment of RNA polymerase II (Pol II) via unknown mechanisms. Here, we applied biochemical protein interaction studies together with electron microscopy-based imaging of a reconstituted human preinitiation complex to define the mechanism underlying Pol II recruitment. These data revealed that the herpesviral TBP, encoded by ORF24 in KSHV, makes a direct protein-protein contact with the C-terminal domain of host RNA polymerase II (Pol II), which is a unique feature that functionally distinguishes viral from cellular TBP. The interaction is mediated by the N-terminal domain (NTD) of ORF24 through a conserved motif that is shared in its β- and γ-herpesvirus homologs. Thus, these herpesviruses employ an unprecedented strategy in eukaryotic transcription, wherein promoter recognition and polymerase recruitment are facilitated by a single transcriptional activator with functionally distinct domains.

## Author summary

The β- and γ-herpesviruses mediate their late gene transcription through a set of viral transcriptional activators (vTAs). One of these vTAs, ORF24 in Kaposi's sarcoma-

**Funding:** This work was supported by the National Institutes of Health grants R01AI122528 (to BAG), R01GM63072 (to EN), and R35GM127018 (to EN). EN and BAG were also supported by Howard Hughes Medical Institute. ALD was supported by Damon Runyon Cancer Research Foundation (DRG-2349-18) and the National Science Foundation (DGE 1752814). AFC was supported by the University of California Berkeley Chancellor's Fellowship. The funders had no role in study design, data collection and analysis, decision to publish, or preparation of the manuscript.

**Competing interests:** The authors have declared that no competing interests exist.

associated herpesvirus (KSHV), is a mimic of host TATA-box-binding protein (TBP). We demonstrate that the N-terminal domain of ORF24 and its homologs from other β- and γ-herpesviruses directly bind the unstructured C-terminal domain (CTD) of RNA Pol II. This functionally distinguishes the viral TBP mimic from cellular TBP, which does not bind Pol II. Thus, herpesviruses encode a transcription factor that has the dual ability to directly interact with promoter DNA and the polymerase, a property which is unique in eukaryotic transcription and is conceptually akin to prokaryotic transcription factors.

## Introduction

Eukaryotic transcription begins with the formation of a pre-initiation complex (PIC) at the core promoter, starting with binding of TFIID and deployment of TATA-box-binding protein (TBP) onto the TATA box or pseudo-TATA box region upstream of the transcription start site (TSS). This is followed by recruitment of the other general transcription factors (GTFs), which recruit and position the 12-subunit RNA polymerase II (Pol II) at the core promoter [1]. The largest Pol II subunit, Rpb1, has a low-complexity carboxyl terminal domain (CTD) that in humans is composed of 52 heptapeptide repeats with a consensus sequence of YSPTSPS. The CTD is a regulatory hub responsible for coordinating signals throughout the different stages of transcription and RNA processing [2]. The phosphorylation state of the CTD controls progression through different states of transcription, as well as interactions with other cellular machinery [2, 3]. Pol II with a hypophosphorylated CTD is recruited into the PIC [4], and phosphorylation signals release from the PIC into an elongating complex.

DNA viruses hijack the host transcriptional machinery to direct their own gene expression. Given that the mechanisms governing transcription from viral promoters are often similar to those at host promoters, viruses have been invaluable models for understanding transcription complex assembly and regulation [5–7]. A conserved feature of double-stranded DNA (dsDNA) viruses is the temporal cascade of gene expression that begins with the expression of two classes of early genes, followed by viral DNA replication, and ending with the expression of late genes. In the β- and γ-herpesviruses, immediate early and early genes are transcribed in a manner similar to host genes. In contrast, the mechanism underlying the regulation of late gene transcription remains poorly understood yet is known to be distinct from host and early viral gene transcription.

Late gene transcription in the β- and γ-herpesviruses is divergent from that of the α-herpesviruses and has a number of unique features. First, β/γ late gene transcription is regulated in part by a core promoter sequence 12–15 base pairs in length that has a TATT motif followed by a RVNYS motif in lieu of the canonical TATA box found in early viral promoters and in some cellular promoters [8–11]. Additionally, late gene expression requires at least six viral proteins, called viral transcriptional activators (vTAs), which form a complex at late gene promoters [12–17]. Little is known about the functional role these vTAs play in late gene transcription.

The best studied protein in the vTA complex is a virally encoded TBP mimic (vTBP), encoded by ORF24 in the γ-herpesvirus Kaposi's sarcoma-associated herpesvirus (KSHV). ORF24 is predicted to have a TBP-like domain in the central portion of the protein, which was identified through an *in silico* protein fold threading analysis performed with BcRF1, the homolog of ORF24 from Epstein-Barr virus [18]. Indeed, ORF24 replaces TBP at late gene promoters during infection, and a virus with mutations in the predicted DNA-binding residues in the TBP-like domain of ORF24 is unable to transcribe late genes [19]. Thus, β- and γ-

herpesviruses encode their own vTBP, which promotes efficient transcription from a distinct set of late gene promoters.

While both cellular TBP and ORF24 (vTBP) bind DNA, co-immunoprecipitation experiments revealed that vTBP additionally interacts with Pol II in cells, although the mechanistic basis of this interaction remains unknown, including whether it is direct or bridged by other cellular cofactors. It is notable that a direct interaction with one or more Pol II subunits would be a feature unique to vTBP [12, 19], as recruitment of Pol II to promoters is mediated by TFIIB instead of a direct protein-protein interaction between TBP and Pol II in host transcription [20, 21]. While other viral proteins have been shown to directly or indirectly bind the Pol II CTD [22–24], none are thought to also bind promoter DNA. In contrast, prokaryotic transcription is dependent on sigma factors that facilitate both promoter selection and polymerase recruitment [25]. We were therefore intrigued by the possibility that vTBP is a unique bifunctional eukaryotic transcriptional activator and sought to understand the basis of its ability to recruit Pol II.

Here, we refined the Pol II interaction domain for multiple ORF24 β- and γ-herpesvirus homologs to show a high degree of functional conservation and reveal that ORF24 makes a direct protein-protein contact with the CTD of Pol II. Using an *in vitro* reconstituted PIC assembly assay coupled with negative stain electron microscopy and pulldown experiments, we determined that ORF24-NTD directly binds four unphosphorylated heptapeptide repeats of the Pol II CTD. We conclude that vTBP is a fundamentally unique protein when compared to other eukaryotic Pol II-interacting proteins, as it both directly interacts with the Pol II CTD and binds the core late gene promotor to coordinate late gene expression.

## Results

### A leucine-rich motif is necessary for interaction with RNA polymerase II across β- and γ-herpesviruses

We previously revealed that KSHV ORF24 co-immunoprecipitates with Pol II in cells in a manner dependent on three conserved leucine residues (L73-75; the RLLLG motif) in the N-terminus of ORF24 [19]. β- and γ-herpesviral homologs of ORF24 from murine gammaherpesvirus 68 (MHV68; mu24), Epstein-Barr virus (EBV; BcRF1), and human cytomegalovirus (HCMV; UL87) have also been reported to interact with Pol II in cells [12, 19, 26]. While this interaction requires no other viral proteins, how it is orchestrated remains a central open question. For example, it is unknown whether Pol II binding occurs through an ORF24 domain separable from the region required for binding the ORF34 vTA (which links ORF24 to the rest of the late gene transcription complex [14]) or the region required for binding to promoter DNA. Furthermore, it is not known whether ORF24 binds Pol II directly or indirectly via bridging cellular factors, as is the case for all other eukaryotic promoter DNA binding transcription factors.

The RLLLG motif is well-conserved in all β- and γ-herpesvirus homologs, despite overall poor sequence identity (**S1 Fig**), suggesting that this N-terminal region of ORF24 homologs may also be necessary for Pol II recruitment. To test if the three conserved leucine residues are involved in the homolog-Pol II interactions, we generated full-length wild-type or triple leucine mutants (3L_A) of ORF24, BcRF1, mu24, and UL87 with C-terminal Strep tags, transiently transfected plasmids encoding these constructs into HEK293T cells, and affinity purified using StrepTactinXT beads. Similar to ORF24, mutation of the RLLLG motif ablated the interactions of all homologs with Pol II (**Fig 1**), suggesting that vTBPs interact with Pol II through their respective N-terminal domains in a manner dependent upon this highly conserved patch of residues.

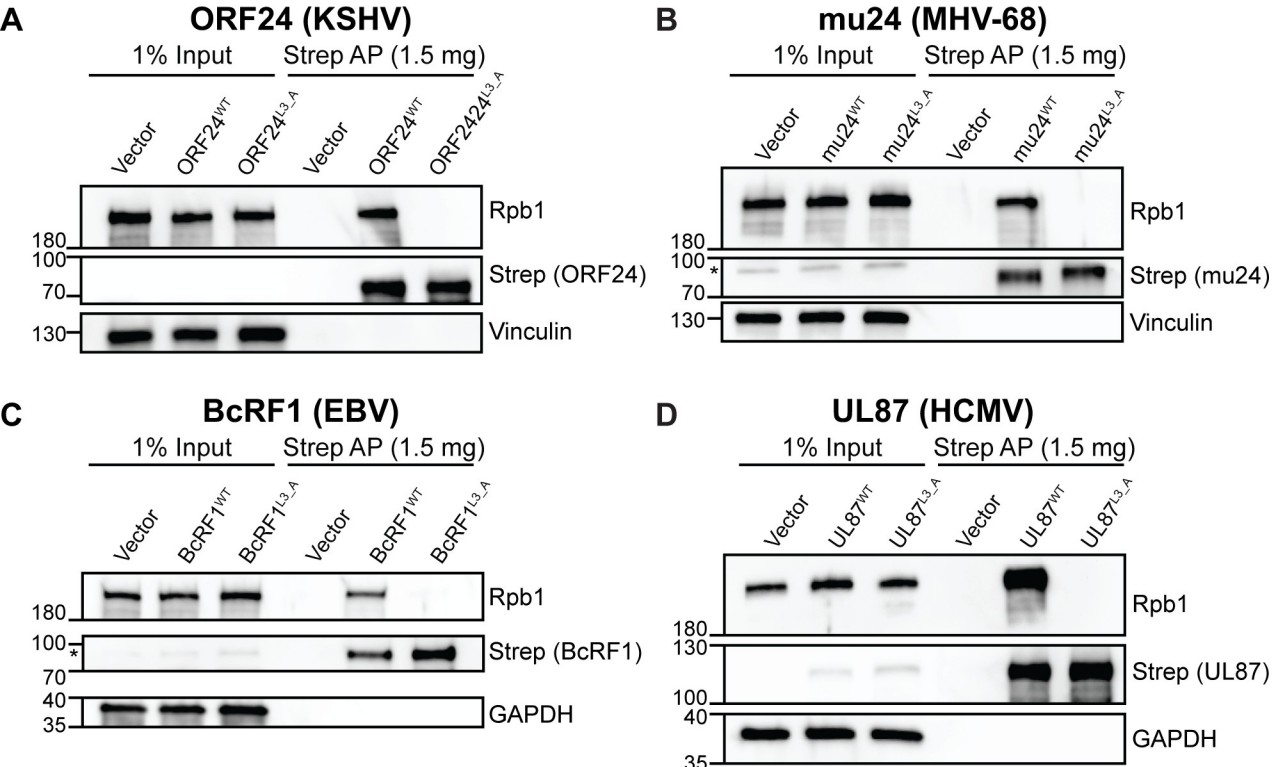

**Fig 1. The RLLLG motif in ORF24 homologs from other β- and γ-herpesviruses is required for interaction with Pol II.** Full-length WT or 3L_A mutants, which mutates the conserved RLLLG motif, of Strep-tagged ORF24 (A) or homologs from MHV68 (mu24) (B), EBV (BcRF1) (C), and HCMV (UL87) (D) were transiently transfected into HEK293T cells then co-affinity purified (AP) with StrepTactinXT beads followed by western blotting. (*) indicates the presence of a non-specific band seen while using the anti-Strep antibody.

### The N-terminal domain of ORF24 is sufficient for interaction with Pol II

Next, to test if the N-terminal domain is sufficient for the Pol II interaction, we constructed a series of N-terminal ORF24 fragments to identify a minimal region of the protein that retained the Pol II interaction (**Fig 2A**). Plasmids encoding full-length or truncated FLAG-tagged ORF24 were transfected into HEK293T cells and immunoprecipitated using FLAG antibodies to determine which ORF24 segments retained binding to endogenous Pol II. The smallest fragment of ORF24 that retained the Pol II interaction consisted of amino acids (a.a.) 1–201, which we termed the ORF24 N-terminal domain (ORF24-NTD) (**Fig 2B**). Notably, the remainder of the protein (a.a. 202–752), which includes the vTBP domain and a region known to interact with the vTA ORF34 [14], failed to co-immunoprecipitate Pol II. Thus, ORF24 contains an NTD that is both necessary and sufficient for Pol II binding, which is separable from the other two known functions of the protein.

### Identification of a minimal Pol II-interaction domain in the β- and γ-herpesviruses

We next sought to determine if the NTD of homologs of ORF24 was similarly sufficient for Pol II interaction, with the goal of identifying whether the β- and γ-herpesviruses share a common domain for polymerase recruitment. Based on the boundary of the ORF24-NTD identified in Fig 2 and sequence alignments (**S1 Fig**), we designed constructs for mu24, BcRF1, and UL87 that encompassed either an analogous NTD, or versions of the homologs where this

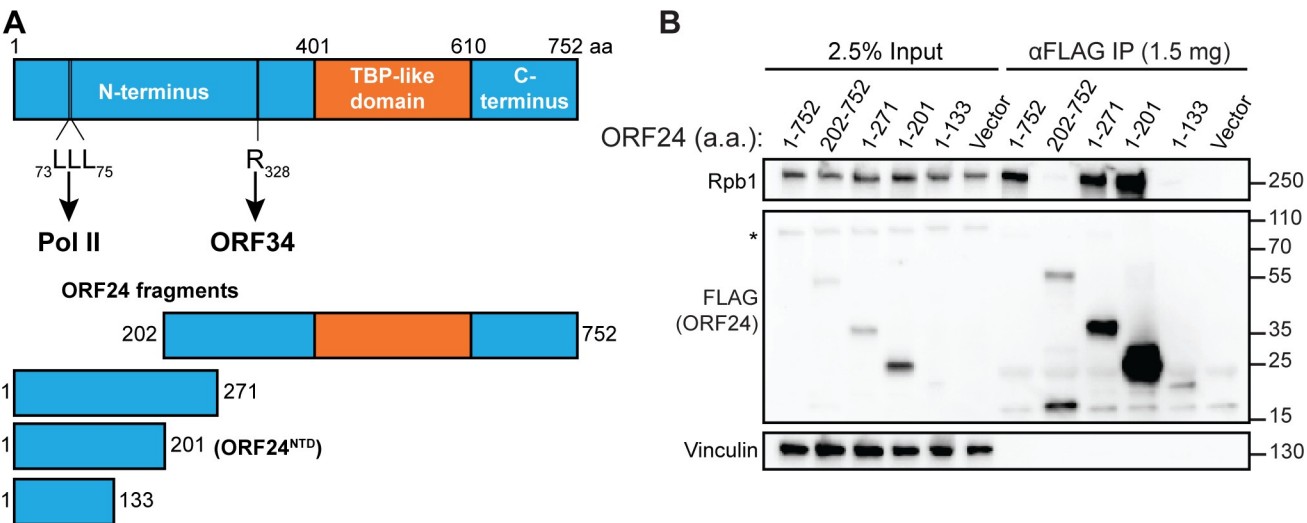

**Fig 2. The N-terminal domain of ORF24 (ORF24-NTD) binds Pol II.** (A) Schematic of constructs used to identify a minimal N-terminal domain of ORF24 showing the predicted boundaries for the N-terminal domain, the TBP-like domain, and the C-terminal domain, including residues known to be required for Pol II binding (amino acids 73–75) and interaction with ORF34 (amino acid 328). (B) HEK293T cells were transiently transfected with full-length or truncated FLAG-tagged ORF24 and co-immunoprecipitated (IP) with anti-FLAG beads followed by western blotting with the indicated antibodies to detect ORF24 and Pol II. (*) indicates the presence of a non-specific band seen while using the anti-Strep antibody.

domain is truncated by ten a.a. or extended by 25 a.a.. Based on isoelectric point (pI) calculations [27], the full-length proteins are predicted to be electropositive, but the N-terminal protein-protein interaction domains are predicted to be electronegative (**S1 Fig**). The BcRF1 NTD appears to be an outlier in terms of overall predicted pI, which may change its physical properties when taken out of context of the full-length protein.

As expected, Strep-tagged ORF24 a.a. 1–201 (ORF24-NTD) or the extended domain containing a.a. 1–226 both interacted with Pol II in Strep affinity purifications from whole cell lysate (**Fig 3A**). Notably, reducing the ORF24-NTD by even ten a.a. eliminated its ability to interact with Pol II, suggesting that ORF24-NTD (a.a. 1–201) is, or is nearly, the minimal domain for the Pol II interaction (**Fig 3A**). To test whether ORF24 1–191, which is expressed worse than ORF24 1–201 or ORF24 1–226, maintains a weak interaction with Pol II that we were not observing due to the dynamic range of our western blots, we reduced expression of the ORF24 1–201 and 1–226 constructs so that ORF24 1–191 had the highest expression (**S2A Fig**). Even when ORF24 1–191 is the highest expressed construct, no interaction with Pol II can be observed, revealing that further truncation from a.a. 1–201 is deleterious to the ORF24-Pol II interaction.

The mu24 and UL87 proteins showed a similar pattern, in which the predicted domain equivalent to ORF24-NTD was sufficient for Pol II interaction, but further truncation by 10 a.a. eliminated the interaction (**Fig 3B and 3D**). The interaction between mu24 and Pol II was very weak, but reproducible and above background (**S2B Fig**). In contrast, in these experiments, all truncations of BcRF1 failed to interact with Pol II (**Fig 3C**, **S2C Fig**).

The above findings indicated that at least 3 of the vTBP homologs share a common N-terminal domain that, despite substantial sequence divergence, is necessary and sufficient for Pol II binding but distinct from other known functional regions of the proteins. Previous data demonstrated that the full-length vTBP from one virus cannot complement homologs in other herpesviruses [12]. However, we considered the possibility that the specific Pol II recruitment domain might be functionally interchangeable between these vTBP homologs if the primary

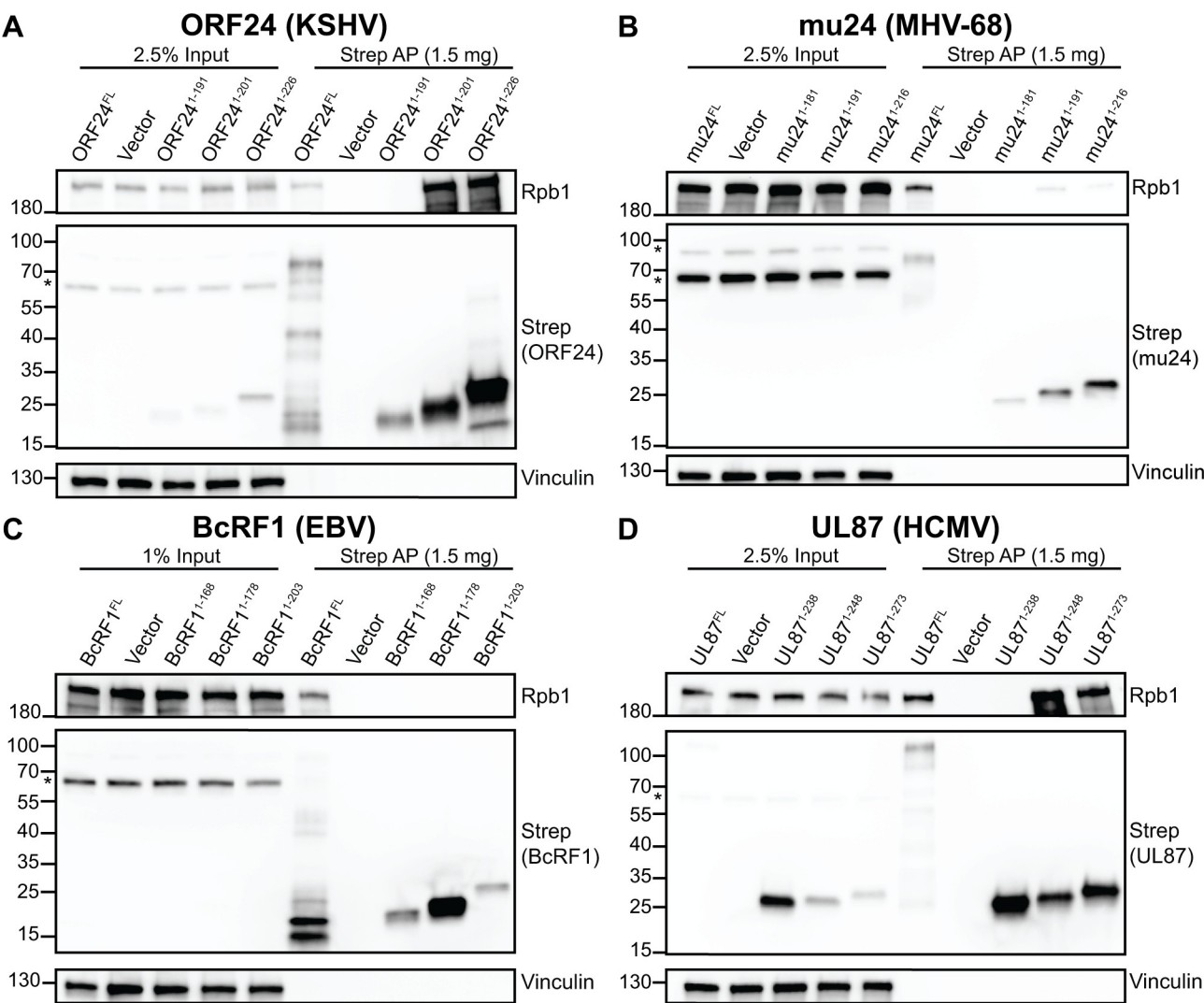

**Fig 3. The N-terminal domain of ORF24 homologs from other β- and γ-herpesviruses is sufficient for interaction with Pol II.** (A-D) Full-length or truncated Strep-tagged constructs of ORF24. (A) or homologs from MHV68 (mu24; B), EBV (BcRF1; C), and HCMV (UL87; D) were transiently transfected into HEK293T cells then co-affinity purified (AP) with StrepTactinXT beads followed by western blotting. (*) indicates the presence of a non-specific band seen while using the anti-Strep antibody.

role of this domain is to bring Pol II to late promoters. To test this, we generated chimeras of ORF24 wherein the ORF24-NTD (a.a. 1–201) was replaced by the minimal NTD of its homologs (**S3A Fig**). These chimeras retain the region of ORF24 that we previously identified as important for the ORF24-ORF34 interaction [14] as well as the ORF24 vTBP domain [18] and C-terminal domain. We noted that the full-length homologs of ORF24 do not interact with KSHV ORF34, despite conservation of an arginine (ORF24 R328) essential for the ORF24-ORF34 interaction [14] (**S3B Fig**). However, each of the NTD chimeras interacted with both Pol II and KSHV ORF34, suggesting that the minimal NTD is sufficiently well-folded when fused to ORF24 a.a. 202–752 (**S3C Fig**). Interestingly, although the minimal domain of BcRF1 alone failed to interact with Pol II by co-IP (**Fig 3D**), it is capable of interaction with Pol II when fused to ORF24 (**S3C Fig**), suggesting that the BcRF1 minimal domain is sufficient for interaction with Pol II, but may have properties not compatible with Pol II interaction when truncated.

We assessed the ability of the NTD chimeras to functionally complement ORF24 using an established transfection-based late gene transcription assay [13–15]. The six vTAs (ORFs 18, 30, 31, 34, 66 and either WT ORF24, its homologs, or chimeras) were co-transfected into HEK293T cells, along with a firefly luciferase reporter controlled by the K8.1 late gene promoter or, as a control, the early ORF57 promoter. A Renilla luciferase reporter was also included to control for transfection efficiency. Consistent with previous observations [12], none of the full-length homologs could functionally complement ORF24 to activate the K8.1 promoter (**S3D Fig**). However, the mu24-ORF24 chimera promoted transcription to levels ~40% that of wild-type ORF24. Interestingly, neither the BcRF1-ORF24 or UL87-ORF24 chimeras were functional for late gene transcription, despite the fact that they could interact with both ORF34 and Pol II (**S3D Fig**). This may suggest that the N-terminal domain has additional functions or interactions beyond polymerase recruitment that mu24 is able to maintain due to sequence similarity to ORF24, or it may simply be that the mu24 fusion (but not the BcRF1 or UL87 fusions) is positioned relative to the promoter and other vTAs in a manner similar enough to ORF24 to be functional. Thus, although the minimal NTD of all vTBPs interacts with Pol II, other contacts or functions may be necessary to successfully promote late gene transcription.

The minimal Pol II interaction domain identified here varies greatly in length (191 a.a. in mu24 vs. 248 a.a. in UL87) and in sequence, as no residues are conserved in both β- and γ-herpesvirus NTDs other than the RLLLG motif (**S1 Fig**). Despite this significant variation, ORF24 and its homologs have evolved a shared mechanism for Pol II recruitment that is primarily mediated by their respective N-terminal domains.

## Negative stain electron microscopy of PICs with GST-ORF24-NTD suggests an interaction with the Pol II stalk

We next sought to determine how ORF24-NTD interacts with Pol II within a minimal PIC. We have as yet been unable to purify full-length ORF24 in sufficient quantity for structural studies, and thus used a construct containing the minimal ORF24-NTD (a.a. 1–201). To generate purified minimal ORF24-NTD, we appended it to an N-terminal glutathione-S-transferase (GST) tag, and achieved robust expression of the protein in *E. coli*, similar to that of GST alone (**Fig 4A**). We confirmed that GST-tagged ORF24-NTD retained the ability to interact with Pol II using a GST pulldown with GST-ORF24-NTD and whole cell lysate from HEK293T cells, suggesting that recombinantly expressed GST-ORF24-NTD is well-folded (**Fig 4B**).

Based on our current understanding of the vPIC, we would expect ORF24 to bind both the TATT promoter element and Pol II, and for the vTAs and some subset of host GTFs to be present. However, we cannot yet assemble a reconstituted vTA complex *in vitro* or purify full length ORF24. Instead, we assembled a PIC using cellular TBP together with the minimal set of cellular GTFs for promoter assembly, to which we added ORF24-NTD. We used a streptavidin-immobilized DNA scaffold based on the super core promoter element (SCP) [28], which contains the TATA, BRE, and INR core promoter elements (**Fig 4C**). Human TBP, TFIIA, TFIIB, and TFIIF were purified from *E. coli*, and Pol II was immunopurified from HeLa cell extracts [29]. We expected that TBP binding to the TATA motif would initiate formation of the PIC with binding of TFIIA and TFIIB, followed by Pol II and TFIIF recruitment, and then binding of GST-ORF24-NTD. Therefore, any additional density beyond that seen in the well-characterized minimal PIC (DNA/TBP/TFIIA/TFIIB/Pol II/TFIIF) [28] could be attributed to ORF24-NTD.

The minimal PIC was assembled by sequentially incubating the DNA scaffold with TBP, TFIIA and TFIIB, Pol II and TFIIF, and GST-ORF24-NTD, followed by immobilization on

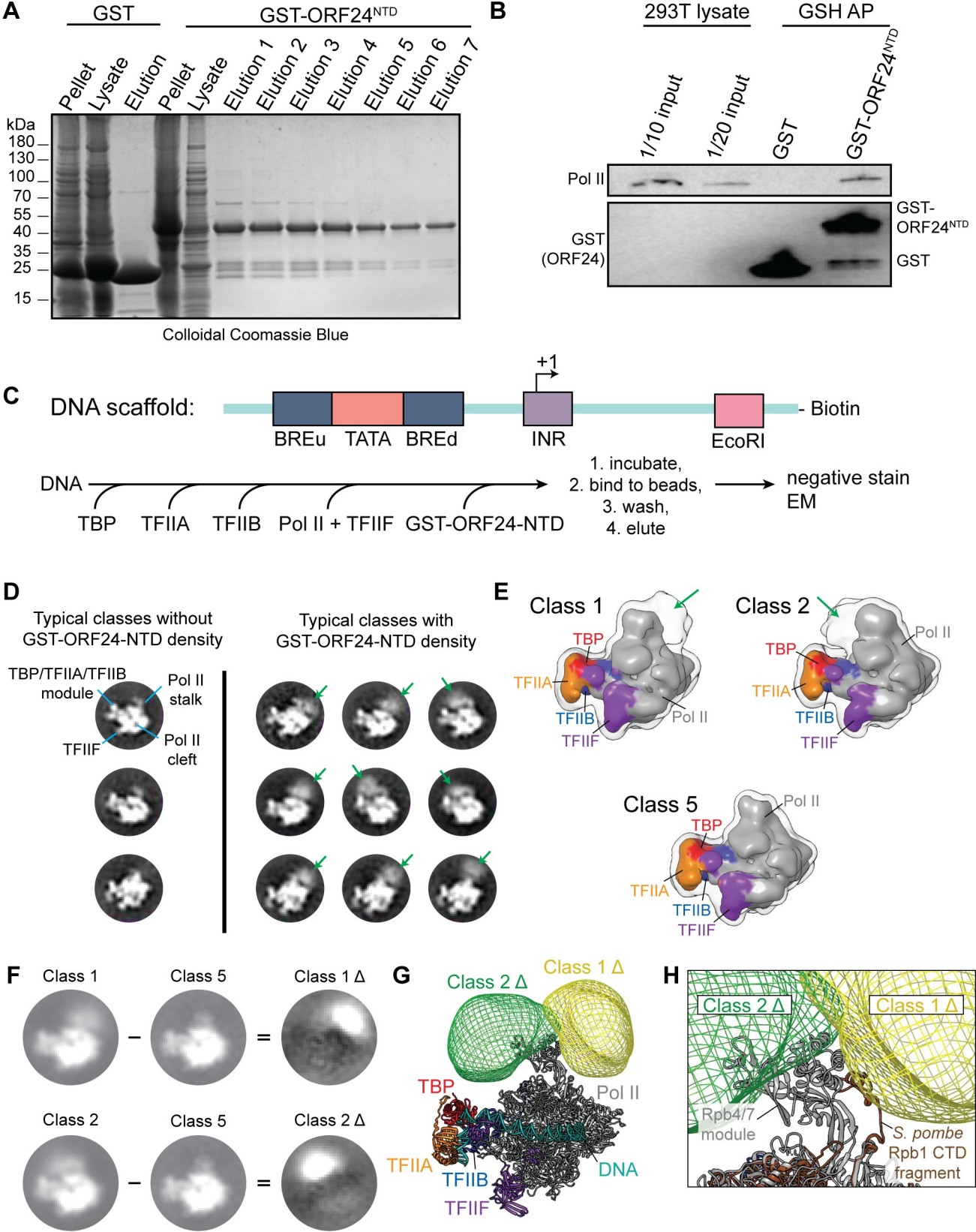

**Fig 4. ORF24-NTD binds Pol II in minimal PICs.** (A) Colloidal Coomassie gel demonstrating that GST and GST-ORF24-NTD can be recombinantly expressed in E. coli and purified by glutathione sepharose purification. (B) GST or GST-ORF24-NTD was incubated in HEK293T whole cell lysate, then subjected to affinity purification using glutathione magnetic beads (GSH AP) followed by western blotting. (C) Sequential reconstitution strategy for a minimal PIC containing GST-ORF24-NTD. (D) Representative reference-free two-dimensional class averages of negatively stained minimal PICs (TBP/TFIIA/TFIIB/TFIIF/Pol II/DNA) assembled in the presence of GST-ORF24-NTD. Three classes on the left show different views of the minimal PIC alone, with the class average in the upper-left annotated with the main features of a minimal PIC particle. The nine class averages on the right show diffuse density in various positions around the Pol II stalk attributed to bound GST-ORF24-NTD (green arrows). (E) Representative three-dimensional class averages of negatively stained minimal PICs assembled in the presence of GST-ORF24-NTD. Classes 1 and 2 exhibit two major areas occupied by bound GST-ORF24-NTD proximal to the Pol II stalk, while class 5 does not exhibit any such density near the Pol II stalk. Solid surfaces are colored by subunit, while a lower intensity iso-surface is shown in transparency to reveal the weaker density attributed to bound GST-ORF24-NTD (green arrows). (F) Difference mapping of the densities attributed to bound GST-ORF24-NTD. Shown on the left are two-dimensional projections of class 1 (top) and 2 (bottom) from (E), and on the right are the difference maps, called "Class 1 Δ" and "Class 2 Δ", calculated by subtracting Class 5 from each of the respective classes. (G) Three-dimensional difference maps corresponding to the extra density attributed to bound GST-ORF24-NTD, mapped onto the structure of the minimal PIC (PDB 5IYA). (H) Zoomed in view of (G) with the structure of Schizosaccharomyces pombe Rpb1 (PDB 3H0G) superposed onto the human structure to show the location of the beginning portion of the Rpb1 CTD within the Pol II stalk. Note that only the very N-terminal portion of the Rpb1 CTD is visible in this structure, with >450 amino acids following this sequence in the CTD of human Rpb1.

streptavidin-coated beads, and finally washing and elution from the beads (**Fig 4C**). We performed single particle negative stain EM of this assembled complex containing GST-ORF24-NTD and searched for particles that displayed extra density beyond the well-defined minimal PIC structure lacking GST-ORF24-NTD. A total of 79,381 PIC particle images were analyzed by two-dimensional and three-dimensional classification, and approximately 25% showed clear extra density in the resulting class averages (**Fig 4D** and **S4 Fig**). One of the three-dimensional classes (Class 5) comprises the majority of the particles and corresponds well to the negative stain EM reconstruction of the human minimal PIC [28], while two classes (Class 1 and 2) exhibit extra density on opposing faces of the Pol II stalk that cannot be attributed to a component of the minimal PIC. Thus, Class 5 was used to subtract the density corresponding to the minimal PIC from Classes 1 and 2, resulting in difference maps that clearly show the region occupied by bound GST-ORF24-NTD (**Fig 4F**). Given the high conformational flexibility of the CTD, we interpret Classes 1 and 2 to represent GST-ORF24-NTD bound in two different orientations on the PIC.

By superimposing the three-dimensional difference maps representing the extra density attributed to GST-ORF24-NTD onto the cryo-EM structure of the human minimal PIC (**Fig 4G**) [30], it became evident that the ORF24-NTD is flexibly bound to a region of Pol II near the Rpb4/7 stalk module. An inherently flexible domain of Pol II is the CTD of the largest subunit, Rpb1, which contains a linker followed by 52 heptapeptide repeats [2] that is not visualized by EM due to its disordered structure. Docking of the crystal structure of Rpb1 from *Schizosaccharomyces pombe* (PDB 3H0G) [31], in which the structure of the CTD linker region is partially resolved and can be seen to extend along the surface of the stalk module, further indicates that the ORF24-NTD may be interacting with the flexible CTD of Pol II (**Fig 4H**). Together, these observations suggest that the NTD of the ORF24 vTBP interacts directly with one or more Pol II subunits, which would be a unique feature distinct from other characterized eukaryotic viral or cellular transcription factors.

## ORF24-NTD binds the CTD repeats of Rpb1

Based on the EM results, we hypothesized that GST-ORF24-NTD could be interacting with either Rpb4/7, the Rpb1 linker, or the Rpb1 CTD heptapeptide repeats. We therefore assessed whether recombinantly purified versions of each of these factors bound purified ORF24-NTD. We first performed GST-pulldown assays with GST-tagged Rpb1 CTD or Rpb1 linker and maltose-binding protein (MBP)-tagged ORF24-NTD. As a control, we also purified an MBP-tagged mutant version of ORF24-NTD, in which the three conserved leucines at positions

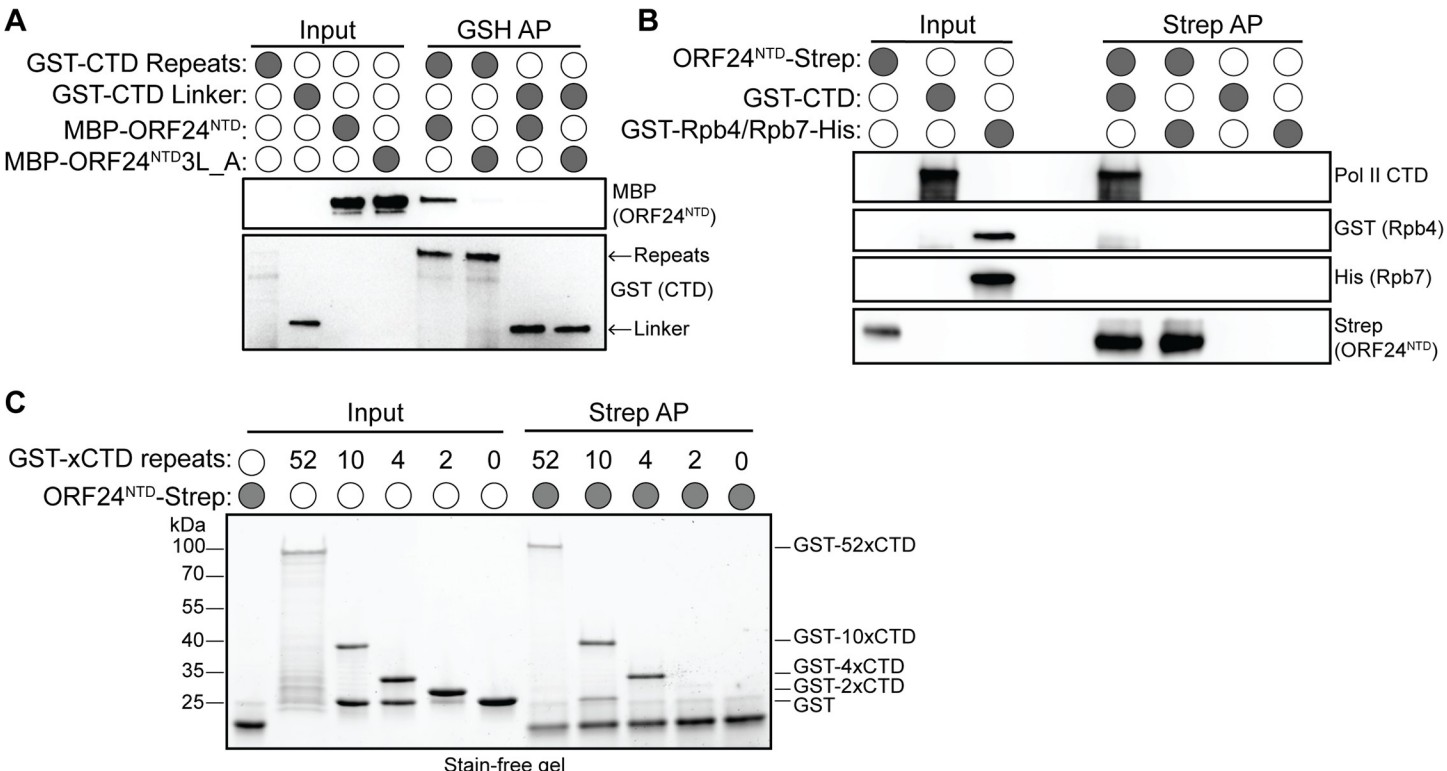

**Fig 5. ORF24-NTD directly interacts with four heptapeptide repeats of the Pol II CTD.** (A) Recombinantly purified GST-CTD repeats or the GST-CTD linker were incubated with either purified MBP-ORF24-NTD or MBP-ORF24-NTD-3L_A protein, then subjected to a glutathione pulldown (GSH AP). Samples were resolved by SDS-PAGE and analyzed by western blot. (B) Recombinantly purified GST-CTD repeats or GST-Rpb4/His-Rpb7 heterodimer were incubated with recombinantly purified Strep-tagged ORF24-NTD, then subjected to a StrepTactinXT pulldown (AP). Samples were resolved by SDS-PAGE and analyzed by western blot. (C) Recombinantly purified full-length GST-CTD repeats (52 repeats) shorter GST-CTD constructs (10, 4, or 2 repeats), or GST itself were incubated with recombinantly purified Strep-tagged ORF24-NTD, then subjected to a StrepTactinXT pulldown (AP). Samples were resolved by SDS-PAGE and stained with colloidal Coomassie.

L73-75 were mutated to alanines (ORF24-NTD-3L_A). This mutation renders the protein unable to interact with Pol II in cells (**Fig 1**, [19]). Notably, WT ORF24-NTD bound to the Rpb1 CTD repeats, but not to the Rpb1 linker region (**Fig 5A**). This interaction was specific and mimicked the results in mammalian cell lysate, as no binding to either CTD fusion was observed with ORF24-NTD-3L_A (**Fig 5A**). We also performed Strep pulldown assays using ORF24-NTD-Strep and the heterodimeric GST-Rpb4/Rpb7-His complex. Again, ORF24-NTD interacted with GST-CTD, but not with GST-Rpb4/Rpb7-His (**Fig 5B**). Thus, recombinant ORF24-NTD directly interacts with recombinant Pol II CTD repeats *in vitro*, and enrichment using tags on either the Pol II CTD or on ORF24 allows for the isolation of the other. The lack of interaction with the CTD linker domain or the Rpb4/7 stalk suggests that the CTD repeats are likely to be the primary point of Pol II contact with ORF24.

The CTD is a key regulatory component of Pol II, responsible for coordinating signals through interactions with multiple transcriptional modulators [2]. CTD binding proteins engage with the heptapeptide repeats in a variety of ways, from recognition of a few residues of a repeat (in the case of kinases that promote phosphorylation at conserved serines, threonines, or tyrosines within a given repeat), to multiple consecutive repeats (in the case of Mediator [32]), to recognition of the intrinsically disordered entire domain (in the case of newly-appreciated phase-separated interactions [33, 34]). To determine which of these types of interactions occur between the CTD and ORF24, we performed Strep pulldown assays with purified

ORF24-NTD-Strep and recombinant GST-CTD constructs containing either 2, 4, 10, or 52 (full length) heptapeptide repeats. The recombinant GST-CTD used in these assays was unphosphorylated, as our previous results demonstrated that ORF24 co-immunoprecipitates only hypophosphorylated Pol II from mammalian cells [19]. Notably, ORF24 interacted with 4x, 10x, and 52xCTD, but failed to interact with either 2xCTD repeats or with control GST (**Fig 5C**). Therefore, similar to the Mediator complex [32], ORF24 likely engages four tandem repeats to interact with the Pol II CTD.

## Discussion

Late gene transcription in the β- and γ-herpesviruses depends on a set of viral transcriptional activators, including a virally encoded mimic of host TBP. Here, we demonstrate that the N-terminal domain (NTD) of the ORF24 vTBP from KSHV recruits Pol II through a direct protein-protein interaction with four heptapeptide repeats of the Pol II C-terminal domain (CTD). Conserved residues in the ORF24-NTD are required for this interaction, suggesting that vTBPs in β- and γ-herpesviruses have evolved a shared strategy for recruitment of Pol II to late gene promoters. Domain swapping experiments between homologs suggest that the primary function of the NTD of vTBP in β- and γ-herpesviruses is to recruit Pol II to the late gene promoter and that the NTD is functionally distinct from the remainder of the protein. Our work conclusively demonstrates that vTBP is unique among eukaryotic transcriptional activators in its ability to simultaneously bind promoter DNA and Pol II, and that this strategy is conserved across the β- and γ-herpesviruses.

Eukaryotic PIC assembly is nucleated by the deployment of TBP onto the promoter DNA by TFIID [35] and is followed by recruitment of other GTFs and Pol II. Unlike the direct interaction that occurs between the ORF24 vTBP and Pol II, the interaction between cellular TBP and Pol II is bridged through GTFs. The β- and γ-herpesviruses have thus adopted a solution reminiscent of bacterial sigma factors, wherein vTBP binds promoter DNA while also recruiting the polymerase directly to the promoter [36]. This raises the question of whether vTBP-mediated Pol II recruitment alters the requirement or roles for other GTFs that contact Pol II. Of particular interest is the role or presence of TFIIB, which both interacts with TBP and is necessary for Pol II recruitment early in PIC formation, functions that may not be necessary in the vPIC given the interactions mediated by vTBP [20, 37]. In this regard, it has recently been shown that the levels of some components of the Pol II transcriptional machinery, including TFIIB, are significantly decreased at late times during lytic gammaherpesvirus infection [38]. It is possible that these viruses assemble alternative transcription complexes in part to compensate for reduced availability of key host factors. The role or presence of other GTFs in vPIC assembly, including TFIIE and TFIIH, which are important for promoter opening and phosphorylation of the Pol II CTD, are also unknown [39]. An intriguing possibility is that the other vTAs fulfill some subset of GTF-like functions during formation of the late gene vPIC. In this regard, we have recently demonstrated that the KSHV vTAs ORF30 and ORF66 are essential for stable binding of ORF24 to the K8.1 late gene promoter [15]. Determining which of the canonical GTFs are engaged in the vPIC will be key to understanding how this unique hybrid virus-host complex activates late gene transcription.

Our work reveals that ORF24 directly binds hypophosphorylated CTD repeats, consistent with a role facilitating viral PIC formation on late gene promoters. CTD-interacting proteins exhibit exceptional diversity in their strategy for recognition of the CTD and in their preference for phosphorylation [40]. Relatively few proteins interact with hypophosphorylated CTD; these proteins (TFIID, TFIIE, TFIIF, and Mediator) are all involved in PIC formation, as phosphorylation of the CTD results in release from the PIC into elongating complexes [41]. The

best characterized hypophosphorylated CTD interacting protein is the Mediator coactivator complex [42]. Mediator is a multi-subunit complex, and its interactions with the Pol II CTD are extensive, involving numerous Mediator subunits [32, 43–45]. A crystal structure of the Mediator head module with the CTD revealed coordination of nearly four CTD repeats [32]. Given the requirement for ORF24-NTD to bind at least four CTD repeats, we are intrigued by the possibility that vTBP functionally or structurally mimics the Mediator head module. One striking difference is the small size of the ORF24-NTD domain that nevertheless efficiently binds the CTD. Mediator is thought to transmit signals from transcription factors bound at regulatory elements to the basal transcriptional machinery [46]. Late gene promoters have exceedingly minimal promoters and lack identified enhancer elements [8]; thus, it is unclear if a Mediator-like function is required for transcription, or if recruitment of Pol II to the promoter is sufficient for transcription. The other vTAs may play a role in bridging currently unidentified enhancers or other elements and may communicate this information to Pol II through the ORF24-ORF34 interaction.

A chimeric vTBP in which the minimal Pol II-interacting domain from ORF24 is replaced with that of mu24 is able to recruit Pol II and maintain interactions with other KSHV vTAs in order to form the vPIC and facilitate transcription. This is the first example of functional interchangeability of any subcomponent of the vTA complex. However, the role of vTBP in late gene transcription extends beyond polymerase recruitment, as vTBP also recruits the remainder of the vTAs to the promoter through protein-protein interactions while also directly binding the late gene promoter DNA.

In summary, ORF24 is a viral transcriptional activator that replaces TBP at late gene promoters and directly recruits Pol II to transcribe viral late genes. That ORF24 binds both the unphosphorylated CTD of Pol II and promoter DNA makes it unique among known eukaryotic CTD-interacting proteins. Since these two functions of ORF24 are genetically separable, they can be characterized independently of one another. Future work to gain atomic-level insight into how ORF24-NTD coordinates the CTD repeats will advance our understanding of its remarkable role in the regulation of late gene transcription in the β- and γ-herpesviruses.

## Materials and methods

### Plasmids

All primer sequences are listed in **S1 Table**. All plasmids used in this study have been deposited to Addgene. The following ORF24 fragments: residues 1–201 (ORF24-NTD) (Addgene #138420), residues 1–271 (Addgene #138421), and residues 1–133 (Addgene #138422) were PCR amplified from pcDNA4/TO-ORF24-3xFLAG [19] (Addgene #138423) with primers to introduce BamHI and NotI sites and cloned into pcDNA4/TO-3xFLAG (C-terminal tag) using T4 DNA ligase (New England Biolabs). pcDNA4/TO-ORF24 202-752-3xFLAG (Addgene #138424) was generated by inverse site-directed mutagenesis with Phusion DNA polymerase (New England Biolabs) using pcDNA4/TO-ORF24-3xFLAG as a PCR template. PCR products from inverse PCR were DpnI treated, then ligated using T4 PNK and T4 DNA ligase (New England Biolabs).

To generate the plasmid for GST-ORF24-NTD expression (Addgene #138464), ORF24-NTD (residues 1–201) was PCR amplified from pcDNA4/TO-ORF24-3xFLAG with primers to introduce BamHI and NotI sites and cloned into pGEX4T1 using T4 DNA ligase. pGEX4T1 encodes an N-terminal GST tag followed by a thrombin cleavage site. To generate the plasmid for MBP-ORF24-NTD WT (Addgene #138465) and 3L_A (Addgene #138466) expression, ORF24-NTD was PCR amplified from pcDNA4/TO-ORF24-3xFLAG WT (Addgene #138423) or 3L_A (Addgene #138425) [19] with primers to introduce SacI and

BamHI sites and cloned into pMAL-c2X using T4 DNA ligase. pMAL-c2X encodes an N-terminal MBP tag and the plasmids were cloned to express ORF24-NTD with a noncleavable MBP tag. ORF24-NTD was PCR amplified from pcDNA4/TO-ORF24-3xFLAG WT and cloned into the KpnI and EcoRI sites of plasmid p6H-SUMO3 using InFusion. A C-terminal Strep tag on ORF24-NTD was added by inverse PCR to generate p6H-SUMO3-ORF24-NTD--Strep (Addgene #138467).

To make the plasmid for GST-Rpb1-linker expression (Addgene #138468), the linker region of Rpb1 (a.a. 1460–1585) was PCR amplified from HEK293T cDNA with primers to introduce BamHI and NotI sites and cloned into pGEX4T1 using T4 DNA ligase. Rpb4 was PCR-amplified from HEK293T cDNA with primers to introduce a BamHI site and cloned into pGEX4T1 using InFusion. The N-terminal BamHI site was regenerated to keep Rpb4 in the same reading frame as the GST fusion tag. Rpb7 was PCR amplified from HEK293T cDNA with primers to introduce a Shine-Delgarno sequence and a C-terminal 6x-His tag and cloned into the EcoRI and NotI sites of pGEX4T1-Rpb4 using T4 DNA ligase to generate pGEX4T1-Rpb4/7 (Addgene #138484).

The 2x and 4x CTD repeat inserts were ordered as a pair of oligonucleotides from Integrated DNA Technologies (IDT), and the primers were annealed by cooling from 90˚C to room temperature in a water bath. The 10x CTD repeat insert was ordered as a synthesized gene block from IDT (S2 Table). All CTD inserts were cloned into the BamHI and NotI sites of pQLink-GST using InFusion cloning (Addgene #138470–138472) (Clontech). pQLink-GST encodes an N-terminal GST tag followed by a TEV protease cleavage site.

Full-length ORF24 with a C-terminal Strep tag (pcDNA4/TO-ORF24-2xStrep) (Addgene plasmid #129742) was previously described [13]. Full-length ORF24 with the 3L_A mutation was subcloned from pcDNA4/TO-ORF24 3L_A-3xFLAG (Addgene #138425) into the BamHI and XhoI sites of pcDNA4/TO-2xStrep (C-terminal tag) using InFusion cloning (Addgene #138440). Full-length UL87 was PCR amplified from HCMV Towne BAC DNA with primers to introduce EcoRI and XhoI sites and cloned into pcDNA4/TO-2xStrep (C-terminal tag) using T4 DNA ligase (Addgene #138434). Full-length mu24 was PCR amplified from MHV68-infected 3T3 cell cDNA with primers to introduce BamHI and NotI sites and cloned into pcDNA4/TO-2xStrep (C-terminal tag) using T4 DNA ligase (Addgene #138435). Full-length BcRF1 was PCR amplified from pcDNA4/TO-BcRF1-3xFLAG [19] and cloned into the BamHI and XhoI sites of pcDNA4/TO-2xStrep (C-terminal tag) using InFusion cloning (Addgene #138436). Mutations of the RLLLG motif in UL87, mu24, and BcRF1 (3L_A mutations) (Addgene #138437–138439) were generated using inverse PCR site-directed mutagenesis.

The minimal domains, minimal domain—10 a.a., and minimal domain + 25 a.a. for ORF24, mu24, BcRF1, and UL87 were PCR amplified from these plasmids and cloned into BamHI/XhoI-cut pcDNA4/TO-2xStrep (C-terminal tag) using InFusion cloning (Addgene #138441–138452). Chimeras of the minimal domain (NTD) ORF24 homologs with ORF24 202–752 were generated using two-insert InFusion cloning (Addgene #138453–138455) into BamHI/XhoI-cut pcDNA4/TO-2xStrep (C-terminal tag).

Plasmid K8.1 Pr pGL4.16 (Addgene plasmid #120377) contains the minimal K8.1 promoter and ORF57 Pr pGL4.16 (Addgene plasmid #120378) contains a minimal ORF57 early gene promoter and have been described previously [13]. Plasmids pcDNA4/TO-ORF18-2xStrep (Addgene plasmid #120372), pcDNA4/TO-ORF24-2xStrep (Addgene plasmid #129742), pcDNA4/TO-ORF30-2xStrep (Addgene plasmid #129743), pcDNA4/TO-ORF31-2xStrep (Addgene plasmid #129744), pcDNA4/TO-2xStrep-ORF34 (Addgene plasmid #120376) have been previously described [13]. Plasmid pRL-TK (Promega) was kindly provided by Dr. Russell Vance.

## Tissue culture and transfections

HEK293T cells (ATCC CRL-3216) were maintained in DMEM supplemented with 10% FBS (Seradigm). For DNA transfections, HEK293T cells were plated and transfected after 24 hours at 70% confluency with PolyJet (SignaGen).

## Immunoprecipitation and western blotting

Cell lysates were prepared 24 hours after transfection by washing and pelleting cells in cold PBS, then resuspending the pellets in IP lysis buffer [50 mM Tris-HCl pH 7.4, 150 mM NaCl, 1 mM EDTA, 0.5% NP-40, and protease inhibitor (Roche)] and rotating for 30 minutes at 4˚C. Lysates were cleared by centrifugation at 21,000 x $g$ for 10 min, then 1–2 mg (as indicated) of total protein was diluted to 1 mg/mL with IP buffer (50 mM Tris-HCl pH 7.4, 150 mM NaCl) and incubated with pre-washed M2 α-FLAG magnetic beads (Sigma) or MagStrep "type3" XT beads (IBA) overnight. Beads were washed 3x for 5 min each with IP wash buffer (50 mM Tris-HCl pH 7.4, 150 mM NaCl, 0.05% NP-40) and eluted by boiling in 2x Laemmli sample buffer (BioRad).

Lysates and elutions were resolved by SDS-PAGE and analyzed by western blot in TBST (Tris-buffered saline, 0.2% Tween 20) using the following primary antibodies: rabbit α-FLAG (Sigma, 1:2500); Strep-HRP (Millipore, 1:2500); rabbit α-Pol II clone N20 (Santa Cruz, 1:2500); mouse α-GST clone 8–326 (Pierce, 1:2000); mouse α-MBP (NEB, 1:10000); mouse α-Pol II CTD clone 8WG16 (Abcam, 1:1000), rabbit α-Vinculin (Abcam, 1:1000), mouse anti-GAPDH (1:1,000; Abcam). Following incubation with primary antibodies, the membranes were washed with TBST and incubated with the appropriate secondary antibody. The secondary antibodies used were the following: goat α-mouse-HRP (Southern Biotech, 1:5000) and goat α-rabbit-HRP (Southern Biotech, 1:5000).

## Protein expression and purification

**GST-ORF24-NTD, GST-CTD linker, and GST.** Proteins were expressed in Rosetta 2 cells (EMD Millipore) grown in LB at 37˚C and induced with 0.5 mM IPTG at an $OD_{600}$ of 0.7 for 16 hours at 18˚C. The cells were harvested by centrifugation at 6500 x $g$ for 10 minutes. The cell pellets were either frozen or immediately resuspended in lysis buffer [50 mM HEPES, pH 7.4, 200 mM NaCl, 1 mM EDTA, 1 mM DTT, protease inhibitors (Roche)] and lysed by sonication. The insoluble fraction was removed by centrifugation at 21,000 x $g$ for 30 minutes. GST-OR24-NTD, GST-CTD linker, and GST were purified on Glutathione Sepharose (GE Healthcare) by batch purification. The proteins were eluted in wash buffer (50 mM HEPES, pH 7.4, 200 mM NaCl, 1 mM EDTA, 1 mM DTT) containing 10 mM reduced glutathione and dialyzed into storage buffer (50 mM HEPES, pH 7.4, 200 mM NaCl, 1 mM EDTA, 1 mM DTT, 10% glycerol].

**GST-xCTD repeats.** GST-2xCTD repeats, GST-4xCTD repeats, and GST-10xCTD repeats were expressed in BL21 Star (DE3) cells grown in Overnight Express Instant TB Medium (EMD Millipore) at 37˚C and induced at an $OD_{600}$ of 1.0 by decreasing the temperature to 18˚C and growing for an additional 16 hours. The cells were harvested by centrifugation at 6500 x g for 10 minutes. The cell pellets were either frozen or immediately resuspended in lysis buffer [50 mM HEPES, pH 7.4, 300 mM NaCl, 5 mM DTT, 5% glycerol, protease inhibitors (Roche)] and lysed by sonication. The insoluble fraction was removed by centrifugation at 50,000 x $g$ for 30 minutes. The proteins were purified as described above and eluted in wash buffer (50 mM HEPES, pH 7.4, 300 mM NaCl, 5 mM DTT, 5% glycerol) containing 10 mM reduced glutathione.

**MBP-ORF24-NTD.** The protein was expressed in Rosetta 2 cells grown in LB at 37°C and induced with 0.5 mM IPTG at an $OD_{600}$ of 0.7 for 16 hours at 18°C. The cells were harvested by centrifugation at 6500 x *g* for 10 minutes. The cell pellets were either frozen or immediately resuspended in lysis buffer [50 mM HEPES, pH 7.4, 200 mM NaCl, 1 mM EDTA, 1 mM DTT, protease inhibitors (Roche)] and lysed by sonication. The insoluble fraction was removed by centrifugation at 21,000 x *g* for 30 minutes. MBP-ORF24-NTD was purified by gravity column chromatography with Amylose Resin (New England Biolabs). The protein was eluted in wash buffer (50 mM HEPES, pH 7.4, 200 mM NaCl, 1 mM EDTA, 1 mM DTT) containing 10 mM maltose and dialyzed into storage buffer (50 mM HEPES pH 7.4, 200 mM NaCl, 1mM DTT, 10% glycerol).

**ORF24-NTD-Strep.** The protein was expressed in NiCo21 (DE3) cells (New England Biolabs) grown in Overnight Express Instant TB Medium (EMD Millipore) at 37°C and induced at an $OD_{600}$ of 1.0 by decreasing the temperature to 18°C and growing for an additional 16 hours. The cells were harvested by centrifugation at 6,500 x *g* for 10 minutes. The cell pellets were either frozen or immediately resuspended in lysis buffer [100 mM HEPES, pH 7.5, 500 mM NaCl, 0.1% Triton X-100, 10% glycerol, 20 mM imidazole, 1 mM TCEP, protease inhibitors (Roche)] and lysed by sonication. The lysate was cleared by centrifugation at 50,000 x *g* for 30 minutes. The clarified lysate was filtered through a 0.45 μm PES filter (Foxx Life Sciences). The protein was purified on an equilibrated HisTrap (GE Healthcare) and step-eluted in wash buffer (100 mM HEPES, pH 7.5, 500 mM NaCl, 0.1% Triton X-100, 10% glycerol, 1 mM TCEP) containing 500 mM imidazole. The fractions containing 6xHis-SUMO-ORF24-NTD--Strep were pooled. The SUMO tag was cleaved overnight at 4°C with 1 mg of SenP2 protease. Following cleavage of the SUMO tag, ORF24-NTD-Strep was purified on an equilibrated StrepTrap (GE Healthcare) and step-eluted in wash buffer (50 mM Tris-HCl, pH 8.0, 200 mM NaCl, 10% glycerol, 1 mM TCEP) containing 2.5 mM desthiobiotin (IBA). The Strep elution fractions containing ORF24-NTD-Strep were pooled and sized on a Superdex 200 (GE Healthcare) size exclusion chromatography (SEC) column in SEC buffer (20 mM HEPES pH 7.4, 100 mM NaCl, 1 mM TCEP, 5% glycerol). The fractions containing ORF24-NTD-Strep were pooled and concentrated on a 10K Amicon Ultra-15 concentrator (EMD Millipore). Protein aliquots were flash frozen and stored at -70°C.

## Pulldown assays

**GST pulldowns.** To test the interaction between GST-ORF24-NTD and Pol II from mammalian cells, 10 μg GST-ORF24-NTD or 10 μg GST was added to 20 μl of washed Glutathione Magnetic Agarose beads (Pierce) along with 250 μg of HEK293T whole cell lysate. IP wash buffer (50 mM Tris pH 7.4, 150 mM NaCl, 0.05% NP-40, 1 mM EDTA) was added to a final volume of 300 μl. The samples were rotated at 4°C for 1 hour. Following the pulldown, the samples were washed with IP wash buffer three times for 5 minutes each time. After the last wash, the protein was eluted by boiling in 2x Laemmli sample buffer (BioRad). The pulldown to test the interaction between GST-CTD repeats or GST-CTD linker and MBP-ORF24-NTD or MBP-ORF24-NTD 3L_A were performed as described above. The elutions were resolved by SDS-PAGE followed by western blot.

**Strep pulldowns.** To test the interaction between ORF24-NTD and Rpb4/7 or xCTD repeats, 5 μg of ORF24-NTD-Strep was added to 10 μl of washed MagStrep "type 3" XT Beads (IBA) along with 10 μg of GST-Rpb4/Rpb7-6xHis or GST-xCTD repeats. SEC buffer (20 mM HEPES pH 7.4, 100 mM NaCl, 1 mM TCEP, 5% glycerol) including 0.05% NP-40 and 5 mM DTT was added to a final volume of 300 μl. The pulldowns were rotated at 4°C for 1 hour followed by three 5 minute washes. The protein was eluted by boiling in 2x Laemmli sample

buffer. Elutions were resolved by SDS-PAGE on a Stain-free gel (BioRad) followed by western blot.

## Late gene reporter assay

HEK293T cells (1 x $10^6$ cells) were plated in 6-well plates and after 24 h each well was transfected with 900 ng of DNA containing 125 ng each of pcDNA4/TO-ORF18-2xStrep, wild-type pcDNA4/TO-ORF24-2xStrep or a homolog of ORF24, pcDNA4/TO-ORF30-2xStrep, pcDNA4/TO-ORF31-2xStrep, pcDNA4/TO-2xStrep-ORF34, pcDNA4/TO-ORF66-2xStrep (or as a control, 750 ng of empty pcDNA4/TO-2xStrep plasmid in lieu of vTA plasmids), with either K8.1 Pr pGL4.16 or ORF57 Pr pGL4.16, along with 25 ng of pRL-TK as an internal transfection control. After 24 h, cells were rinsed twice with PBS, lysed by rocking for 15 min at room temperature in 500 µL of Passive Lysis Buffer (Promega), and clarified by centrifugation at 21,000 x g for 2 min. 20 µL of the clarified lysate was added in triplicate to a white chimney well microplate (Greiner Bio-One) to measure luminescence on a Tecan M1000 microplate reader using a Dual Luciferase Assay Kit (Promega). The firefly luminescence was normalized to the internal Renilla luciferase control for each transfection. All samples were normalized to the corresponding control containing no vTAs.

## Negative stain electron microscopy

**PIC assembly and purification.** TBP, TFIIA, and TFIIB were recombinantly expressed and purified from *E. coli*. Pol II was immunopurified from HeLa cell nuclear extracts following previously published protocols [29, 47]. The DNA construct was previously described [28] and is an SCP [48] containing a BREu element upstream of the TATA box [21] and an EcoRI restriction enzyme site downstream of the INR element for purification purposes. A biotin tag was coupled to the 5' end of the template strand (Integrated DNA Technologies). The duplex DNA was generated by annealing the single-stranded template strand DNA with equimolar amounts of non-template strand DNA at a final concentration of 50 µM in water. The annealing reaction was carried out by incubating at 98°C for 2 minutes followed by cooling to room temperature at a rate of 1°C per second.

PICs were assembled in assembly buffer (20 mM HEPES, pH 7.9, 0.2 mM EDTA, 10% glycerol, 6 mM $MgCl_2$, 80 mM KCl, 1 mM DTT, 0.05% NP-40). DNA was used as a scaffold and purified TBP, TFIIA, TFIIB, Pol II, TFIIF, and GST-ORF24-NTD were sequentially added into the assembly buffer. Following assembly of the PICs, the reaction was incubated at 28°C for 15 minutes using a 1:10 dilution of magnetic streptavidin T1 beads (Invitrogen), which had been previously equilibrated in assembly buffer. The beads were washed three times with wash buffer (10 mM HEPES, 3% trehalose, 8 mM $MgCl_2$, 100 mM KCl, 1 mM DTT, 0.025% NP-40). The complex was eluted by incubation at 28°C for 1 hour in digestion buffer (10 mM HEPES, pH 7.9, 3% trehalose, 10 mM $MgCl_2$, 50 mM KCl, 1 mM DTT, 0.01% NP-40, 1 unit µL$^{-1}$ EcoRI-HF (New England Biolabs)). After elution, purified PIC was crosslinked on ice in 0.05% glutaraldehyde for 5 minutes then immediately used for EM sample preparation.

**Electron microscopy.** Negative stain samples of PIC were prepared on a 400 mesh copper grid containing a continuous carbon supporting layer. The grid was plasma-cleaned for 10 seconds using a Solarus plasma cleaner (Gatan). An aliquot (3.5 µL) of the purified sample was placed onto the grid and allowed to absorb for 5 minutes at 100% humidity. The grid was then placed sample-side-down on five successive 75 µL drops of 2% (w/v) uranyl formate solution for 10 seconds on each drop followed by blotting to dryness. Data collection was performed on a Tecnai F20 TWIN transmission electron microscope operating at 120 keV at a nominal magnification of 80,000X (1.5 Å/pixel). The data were collected on a 4k X 4k CCD (Gatan) using

low-dose procedures (20 e$^-$ Å$^{-2}$ total dose per exposure), using Leginon software to automatically focus and collect exposure images.

**Image processing.** Data pre-processing was performed using the Appion processing environment [49]. Particles were automatically selected from the micrographs using a difference of Gaussians (DoG) particle picker [50]. The contrast transfer function (CTF) of each micrograph was estimated using both ACE2 and CTFFind [51, 52]. Boxed particle images were extracted using a box size of 256 X 256 pixels from images whose ACE2 confidence value was greater than 0.8, phases were flipped, and images were normalized using the XMIPP to remove pixels which were above or below 4.5$\sigma$ of the mean value [53]. The particle stack was binned by a factor of two and two-dimensional classification was conducted using iterative multireference alignment analysis (MSA-MRA) within the IMAGIC software [54].

**Three-dimensional reconstruction.** Particles belonging to bad two-dimensional classes were thrown out, resulting in a stack of 79,381 single particle images that were used for three-dimensional analysis. Three-dimensional classification was performed within RELION [55], using the cryo-EM structural of a minimal PIC (EMD-2305, [28]), low-pass filtered to 50 Å resolution, as an initial reference, and sorted into 6 classes. The resolution of the reconstructions containing GST-ORF24-NTD were estimated to be ~20 Å.

## Supporting information

**S1 Fig. Multiple sequence alignment of homologs of ORF24 from other β- and γ-herpesviruses.** The conserved triple leucine motif is highlighted in red. The location of truncations for the constructs used in Fig 3 are highlighted in yellow, teal, and purple. Sequences used to construct the alignment are listed in the bottom left. The boundaries for the truncated constructs and their relative predicted isoelectric point based on [27] is shown in the bottom right.
(TIF)

**S2 Fig. The N-terminal domain of ORF24 and mu24, but not BcRF1, is sufficient for interaction with Pol II HEK293T cells were transiently transfected, then co-affinity purified (AP) with StrepTactinXT beads (AP) followed by western blotting.** (*) indicates the presence of a non-specific band seen while using the anti-Strep antibody. (A) Truncated Strep-tagged constructs of ORF24 were used for the transient transfection/AP experiment. In all cases, 5 μg of total plasmid DNA was transfected. For the ORF24 a.a. 1–201 and 1–226 constructs, 1.5 and 1.0 μg of plasmid DNA was used with 3.5 and 4.0 μg of empty vector DNA, respectively. (B) Full-length or truncated Strep-tagged constructs of homologs from MHV68 (mu24) were used for the transient transfection/AP experiment. (C) Full-length or truncated Strep-tagged constructs of homologs from EBV (BcRF1) were used for the transient transfection/AP experiment.
(TIF)

**S3 Fig. vTBP-NTD chimeras can interact with Pol II and a mu24-ORF24 chimera can activate a late gene promoter.** (A) Schematic of construct design for ORF24 chimeras. The ORF24-NTD (a.a. 1–201) was replaced with the experimentally identified minimal domain of mu24, BcRF1, and UL87. These chimeric constructs retain the N-terminal ORF24-ORF34 interaction region, the ORF24 vTBP domain, and ORF24 C-terminal tail. (B) Full-length Strep-tagged homologs of ORF24 were transiently transfected into HEK293T cells along with FLAG-tagged ORF34, then co-affinity purified with StrepTactinXT beads (AP) followed by western blotting. (*) indicates the presence of a non-specific band seen while using the anti-Strep antibody. (C) Full-length Strep-tagged chimeras of ORF24 were transiently transfected into HEK293T cells along with FLAG-tagged ORF34, then co-affinity purified with StrepTactinXT beads (AP) followed by western blotting. (*) indicates the presence of a non-specific

band seen while using the anti-Strep antibody. (D) HEK293T cells were transiently transfected with a pGL4.16 firefly luciferase plasmid driven by either the ORF57 (early gene) or K8.1 (late gene) promoter. Plasmids encoding either ORF24, its homologs, or the chimeras, along with the five remaining KSHV vTAs (ORFs 18, 30, 31, 34, and 66) and a pRL-TK renilla luciferase plasmid (as a transfection control) were also transfected. After 24 h, cell lysates were harvested and luciferase activity was measured. Fold activation was calculated by normalizing to the firefly/renilla signal in the absence of vTAs.
(TIF)

**S4 Fig. Classification of particles from the single particle negative stain EM minimal PIC containing GST-ORF24-NTD.** Two-dimensional projections of the three-dimensional classes resulting from sorting the single particle EM images of negatively stained minimal PICs (TBP/TFIIA/TFIIB/TFIIF/Pol II/DNA) assembled in the presence of GST-ORF24-NTD. The number of particles assigned to each class and the percent of total particles are indicated. Classes 1, 2, and 5 were used for the difference mapping shown in Fig 4.
(TIF)

**S1 Table. List of oligonucleotides used in this study.**
(DOCX)

**S2 Table. Nucleotide sequence of synthetic gene blocks used in this study.**
(DOCX)

## Acknowledgments

We thank Jie Fang for purification of human GTFs and Pol II, and Patricia Grob and Abhiram Chintangal for microscope and computational support, respectively. We are thankful to all members of the Glaunsinger lab, especially Divya Nandakumar, for helpful discussions and suggestions.

## Author Contributions

**Conceptualization:** Angelica F. Castañeda, Allison L. Didychuk, Zoe H. Davis, Eva Nogales, Britt A. Glaunsinger.

**Data curation:** Angelica F. Castañeda, Allison L. Didychuk, Robert K. Louder.

**Formal analysis:** Angelica F. Castañeda, Allison L. Didychuk, Robert K. Louder.

**Funding acquisition:** Eva Nogales, Britt A. Glaunsinger.

**Investigation:** Angelica F. Castañeda, Allison L. Didychuk, Robert K. Louder, Chloe O. McCollum.

**Methodology:** Angelica F. Castañeda, Allison L. Didychuk, Robert K. Louder, Eva Nogales, Britt A. Glaunsinger.

**Project administration:** Angelica F. Castañeda, Allison L. Didychuk, Eva Nogales, Britt A. Glaunsinger.

**Supervision:** Eva Nogales, Britt A. Glaunsinger.

**Validation:** Angelica F. Castañeda, Allison L. Didychuk, Robert K. Louder, Chloe O. McCollum.

**Visualization:** Angelica F. Castañeda, Allison L. Didychuk, Robert K. Louder.

**Writing – original draft:** Angelica F. Castañeda, Allison L. Didychuk.

**Writing – review & editing:** Angelica F. Castañeda, Allison L. Didychuk, Robert K. Louder, Chloe O. McCollum, Zoe H. Davis, Eva Nogales, Britt A. Glaunsinger.

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
