## [Decision Letter · Decision Letter 0]

17 May 2020

Dear Dr. Glaunsinger,

Thank you very much for submitting your manuscript "The gammaherpesviral TATA-box-binding protein directly interacts with the CTD of host RNA Pol II to direct late gene transcription" for consideration at PLOS Pathogens. As with all papers reviewed by the journal, your manuscript was reviewed by members of the editorial board and by several independent reviewers. In light of the reviews (below this email), we would like to invite the resubmission of a significantly-revised version that takes into account the reviewers' comments.

We cannot make any decision about publication until we have seen the revised manuscript and your response to the reviewers' comments. Your revised manuscript is also likely to be sent to reviewers for further evaluation.

Sincerely,

Dirk P. Dittmer, Ph.D.

Associate Editor

PLOS Pathogens

Shou-Jiang Gao

Section Editor

PLOS Pathogens

Kasturi Haldar

Editor-in-Chief

PLOS Pathogens

orcid.org/0000-0001-5065-158X

Michael Malim

Editor-in-Chief

PLOS Pathogens

orcid.org/0000-0002-7699-2064

Reviewer's Responses to Questions

**Part I - Summary**

Reviewer #1: The authors have examined the interaction between vTBP proteins (ORF24, mu24, UL87, BcRFI) expressed in beta and gamma herpesviruses that interact with the late gene TATT promoter element and RNA polymerase II. They conclude that a highly conserved stretch of 5 amino acids (RLLLG) in the N-terminal region of the vTBPs is responsible for binding directly to the CTD of the large subunit of RNA polymerase II when it is hypophosphorylated. The results strongly support this conclusion, but it is this reviewer’s opinion that few important new findings have been made. An important Molecular Cell paper (2015) on ORF24 from the same lab demonstrated an RLLLG-dependent interaction between ORF24 and Pol II. In that paper they demonstrated that the interactions was disrupted when the CTD was phosphorylated. They report here a small, incremental finding that the RLLLG-dependent interaction is with the CTD and that that interaction is at least in part direct.

Specific Comments:

1. There are claims that vTBP is the only know transcription factor that interacts with DNA and Pol II. They say, “We conclude that vTBP is a fundamentally unique protein when compared to other eukaryotic Pol II-interacting proteins, as it both directly interacts with the Pol II CTD and binds promotor DNA to coordinate late gene expression.” It is true that the carboxyl terminal half of the 339 aa human TBP used for structural studies does not appear to interact directly with Pol II. However, TFIIB interacts with the DNA surrounding the TBP binding site and interacts with Pol II. The vTBP is about 3 times larger than TBP with other critical domains not contained within TBP. The goal of bridging the promoter DNA to Pol II is accomplished in the host and virus through the coordinated interactions of DNA binding proteins, Pol II binding proteins and Pol II. There is no reason to expect vPICs would operate using the same connectivity as found the host especially given the large differences in the proteins used.

2. In 2B what are the proteins in the ~100 kDa range in the inputs? Why is the full length protein not expressed well? Why is Pol II pulled down much more efficiently by the full length protein (more than 10 times better)? Are other domains critically important? Pol II CTD should not be the label for the westerns it should be RPB1 or something to indicate it is the large subunit of Pol II. Why are there no phosphorylated forms of the large subunit seen in the input?

3. In 3B NTDs of ORF24 and UL87 do pull our Pol II, but MHV-68 (mu24) and BcRF1 do not. (this is not what was said).

4. It is not clear to me what important was learned from the chimera studies.

5. I could not figure out why negative stain EM was used to examine the interaction of the domain of vTBP with the partial host PIC complex containing TBP. The previous published studies already strongly pointed to the CTD mediating the interaction between vTBP and Pol II. Clearly, this is not a physiologically relevant complex.

In conclusion, the study was performed to high standards, but failed to deliver a satisfactory level of new information.

Reviewer #2: Beta and gamma herpesviruses require at least seven virally encoded proteins to orchestrate transcription of late viral genes. These late gene regulators (or viral transcription activators vTAs) assemble into a viral transcription pre-initiation complex (vPIC) on late gene promoters. One essential component of this complex is the viral TATA-box binding protein (vTBP) that recognizes a TATT element present in late promoters. A previous study from the same group demonstrated the capacity of vTBP homologs to interact with RPB1, the catalytic subunit of RNA polymerase II. The interaction was mediated by three leucine residues (RLLLG motif) in KSHV-TBP (orf24). This presents an interesting model in which vTBPs are directly involved in recruitment of RNA polymerase to transcribe late genes. Such model would be different from transcription of cellular genes in which the host TBP does not directly interact with RNAPII. However, to establish this model two essential questions had to be addressed: does vTBP directly interact with subunits of RNAPII? Which subunit of RNAPII mediates the interaction? In the current manuscript the authors addressed these questions and made the following novel observations:

1) the RLLLG motif, which is conserved among beta and gamma-herpesviruses, mediates the interaction with RPB1.

2) The N-terminal domain of ORF24 (KSHV TBP) homologs is necessary and sufficient for interaction with RPB1.

3) Using single particle negative stain EM, the authors assembled a minimal PIC containing GST-ORF24-NTD and were able to obtain structural information on the NTD of ORF24 bound to RNAPII. This information is crucial as it shows a direct interaction between ORF24 and RNAPII. Two potential interaction interfaces were determined using EM; these are RPB4/7 and RPB1-CTD.

4) Using GST-pulldown assays the authors demonstrated that interaction of ORF24-NTD with CTD requires at least four heptad repeats. No interaction was observed between ORF24-NTD and RPB4/7. Overall, the manuscript is well written, the topic is important, the experiments are well executed and the results are clear and support the conclusions.

Reviewer #3: In this manuscript, Castañeda et al. describes the studies on the interactions between ORF24 of KSHV, a herpesviral TBP and cellular RNA polymerase II (PolII). First, they extended the observation that they previous made on KSHV ORF24 to other herpesival homologues encoded by MHV68, ENV, and HCMV. A conserved leucine-rich motif required for KSHV ORF24 to interact with RNA PolII is also critical for other homologues. Second, they determined the N-terminal domain that contains the leucine-rich motif is sufficient to bind PolII. Third, the in vitro assembled ORF24-PolII minimal PIC complex was examined by EM and the potential regions of PolII interactions were identified. One region, the CTD repeats of Rbp1, was confirmed by GST-pulldown and concluded by the authors to be the primary contact point.

Cellular TBP does not bind PolII directly; instead, the recruitment of PolII is mediated by TFIIB or other GTFs. Therefore, the major contribution from this manuscript is the demonstration of a direct interaction between a herpesviral TBP and PolII. The manuscript is well written and easy to read.

**Part II – Major Issues: Key Experiments Required for Acceptance**

Reviewer #1: (No Response)

Reviewer #2: None

Reviewer #3: 1. There should be a clear description of the rationale using cellular TBP in the in vitro assembly described in Fig. 4 for EM studies, especially because TBP is not part of vPIC as this group previously determined. An explanation of why a TBP-like containing ORF24 and a viral TATA promoter is not used. Has this been attempted but without success? Since only the N-terminal domain of ORF24 without TBP-like domain was used and instead cellular TBP was used in the assembly, a discussion on limitations of the EM result needs to be included. For example, is it expected that the TBP-like domain of ORF24 would occupy the same position as cellular TBP? If not, how this would affect the EM structural results?

2. From Fig.1 to Fig.3, there were several mutant constructs that did not express well or efficiently immunoprecipitated. For example, the 3L mutant of mu24 (Fig. 1), the two mutants (Fig. 2), 202-752 and 1-133, as well as the 1-191 mutant (Fig. 3A), the 1-181 mutant (Fig. 3B) and the 1-203 mutant (Fig. 3C). The conclusion that these mutants do not interact with PolII cannot be drawn unless comparable immunoprecipitation or expression of the mutants is achieved.

**Part III – Minor Issues: Editorial and Data Presentation Modifications**

Reviewer #1: (No Response)

Reviewer #2: Fig 3: In line 172 the authors concluded that all truncations of BcRF1 failed to interact with RNAPII. However, BcRF1(1-203) is expressed at a very low level relative to other BcRF1 mutations. This might explain why the NTD of BcRF1 is insufficient to precipitate RPB1.

S2 Fig: Testing the ability of the NTD chimeras to functionally complement ORF24 was performed using a reporter assay. Perhaps the authors should consider using KSHV infected cells in which expression of the endogenous ORF24 was disrupted.

Fig 4. Would the authors provide explanation on why there are two 3D classes (class 1 and 2) of high density on opposing faces of PolII observed with single particle negative stain EM?

Fig 4F: The authors used class 5 to subtract the density corresponding to the minimal PIC and to identify the region occupied by GST-ORF24-NTD. It is not clear how the authors concluded that class 5 represents minimal PIC that lacks ORF-24-NTD? Is it feasible to compare the density obtained from class 5 molecules with the density of corresponding molecules obtained in He et al (ref 30), assuming availability?

Lines 273/274: To conclude that ORF24 does not interact with RPB4 or RPB7, the authors should consider examining the capacity of full length ORF24 to interact with these two subunits. So far the data suggests that the NTD of ORF24 does not interact with RPB4/7.

Reviewer #3: (No Response)

PLOS authors have the option to publish the peer review history of their article (what does this mean?). If published, this will include your full peer review and any attached files.

Reviewer #1: No

Reviewer #2: No

Reviewer #3: No
---

## [Editor Report · Decision Letter 1]

28 Jul 2020

Dear Dr. Glaunsinger,

We are pleased to inform you that your manuscript 'The gammaherpesviral TATA-box-binding protein directly interacts with the CTD of host RNA Pol II to direct late gene transcription' has been provisionally accepted for publication in PLOS Pathogens.

Best regards,

Dirk P. Dittmer, Ph.D.

Associate Editor

PLOS Pathogens

Shou-Jiang Gao

Section Editor

PLOS Pathogens

Kasturi Haldar

Editor-in-Chief

PLOS Pathogens

orcid.org/0000-0001-5065-158X

Michael Malim

Editor-in-Chief

PLOS Pathogens

orcid.org/0000-0002-7699-2064
---

## [Editor Report · Acceptance letter]

1 Sep 2020

Dear Dr. Glaunsinger,

We are delighted to inform you that your manuscript, "The gammaherpesviral TATA-box-binding protein directly interacts with the CTD of host RNA Pol II to direct late gene transcription," has been formally accepted for publication in PLOS Pathogens.

Best regards,

Kasturi Haldar

Editor-in-Chief

PLOS Pathogens

orcid.org/0000-0001-5065-158X

Michael Malim

Editor-in-Chief

PLOS Pathogens

orcid.org/0000-0002-7699-2064